# Tourist distribution in Northern Mediterranean Basin countries: 2004–2020

**Sabri Öz[1], Adnan Veysel Ertemel**[2]\***, Pınar Başar**[3]**, Cemil Can Çoktuğ**[4]

**1** Faculty of Management, Industrial Policy and Technology Program, Istanbul Commerce University, Istanbul, Turkey, **2** Faculty of Business Administration, Istanbul Technical University, Istanbul, Turkey, **3** Faculty of Management, Business Administration Department, Istanbul Commerce University, Istanbul, Turkey, **4** Social Sciences Institute, Istanbul Commerce University, Istanbul, Turkey

\* ertemelav@itu.edu.tr

**Citation:** Öz S, Ertemel AV, Başar P, Çoktuğ CC (2023) Tourist distribution in Northern Mediterranean Basin countries: 2004–2020. PLoS ONE 18(11): e0293669. https://doi.org/10.1371/journal.pone.0293669

**Data Availability Statement:** All relevant data are within the manuscript and its Supporting Information files.

**Funding:** The author(s) received no specific funding for this work.

## Abstract

### Purpose

The feasibility of measuring the touristic ecosystem in European countries with a Mediterranean coast based on various parameters, including diversity, turnover, and the number of tourists, was investigated in this study. The data from the period between 2004 and 2020 were analyzed.

### Methodology

A distribution analysis of annual tourist gains was conducted, and the distribution of incoming tourists across the countries was examined based on their area, using Atkinson, Theil, and Hoover inequality indices. Secondary data from the World Bank were utilized by the authors for the 13 countries studied. It was suggested by the authors that the Mediterranean region could be analyzed based on factors such as the length of the coast, the number and type of hotel beds, and the volume of coastal tourism. This study can be expressed as a mixed methodology supported by bibliometric analysis.

### Findings

An overall improvement in the distribution of tourists was indicated by the results of the analysis, with the exception of a decline in 2016 and 2020, as confirmed by all three indices. The most significant decline in 2020 was shown by the Hoover Index.

### Originality

This study is a significant contribution to the existing literature, as it is the first to analyze the distribution of tourists considering the Mediterranean Basin coast length and the number of tourists of the illustrated countries, using the Atkinson, Theil, and Hoover inequality indices. The study was deemed original and supported by bibliometric analysis. The results of this study have important managerial implications.

**Competing interests:** The authors have declared that no competing interests exist.

## 1. Introduction

Tourism, which holds significant social, cultural, and economic value for countries, encompasses various forms, such as health, education, beach, entertainment, mental, and nature tourism [1]. It is classified into domestic and foreign tourism based on tourist sources [2]. The growth of health tourism has been particularly impacted by the COVID-19 pandemic, resulting in significant changes in tourism diversity and distribution [3].

This study is distinctive in that it presents the distribution of tourists visiting a specific region, employing inequality indices to analyze the patterns. Notably, the study further stands out by conducting a comparative assessment of the variability in tourist distribution over a specific timeframe, utilizing three different indices. This comparative approach enhances the study's uniqueness, contributing to a more comprehensive understanding of the distribution dynamics of tourist numbers within the region.

In this study, the distribution of tourists among countries and whether the distribution changed during crisis events were examined, without addressing different types of tourism. The study's focus was on Mediterranean countries and countries with borders in the northern region of the Mediterranean. The number of tourists was analyzed based on the land size of the countries, and the study was repeated using three inequality indices: Atkinson, Hoover, and Theil. The regional distribution of tourists was discussed using data obtained from the World Bank database between 2004 and 2020.

The dataset used in this study is derived from the arrivals of tourists in the respective countries. It is important to emphasize that, across all indices employed, the area of the countries is a crucial parameter considered in the analysis.

Some studies have highlighted the favorable situation of Southern European tourism compared to its Northern European counterpart, particularly following the impact of the Covid pandemic [4]. This study specifically examines the distribution of tourists in the South in 2020 under the influence of Covid. Southern European countries, mainly located in the Mediterranean Basin, appear to have experienced relatively less disruption than their northern counterparts. However, it is noteworthy that the deterioration in the number of tourists between countries is evident when analyzing all three indices.

Conversely, Butler's [5] study focused on the potential loss of tourists due to the pandemic-induced tourism break. Interestingly, despite the occurrence of tourist losses, the trend of inequality persists consistently. Even in 2020, where a decline is observed, the pattern tends to revert to its previous trajectory, as demonstrated by similar occurrences of one-year inequality deterioration followed by subsequent downward trends in both 2006 and 2016.

On the other hand, Santos and Moreiara study [6] emphasizes the significance of this research, offering suggestions for revitalizing the tourism sector, particularly in the aftermath of the Covid pandemic.

This study primarily focuses on the Mediterranean basin, a region of exceptional interest in global tourism management. The rationale behind this choice is the extensive attention drawn to Mediterranean basin tourism management on a worldwide scale. A notable example is the book 'Tourism Management,' edited by Medlik, which dedicates a separate section to the Mediterranean basin [7]. Furthermore, the revenue generated from tourism within the Mediterranean basin holds particular significance for the economies of European countries, which are among the world's foremost economic powers [8].

In conclusion, the findings from these various studies underscore the complexity of the tourism sector, with Southern European countries showing resilience amidst the pandemic's impact. Nevertheless, addressing inequality in tourism distribution remains a critical concern,

and the insights provided by this study contribute to understanding the dynamics of the tourism industry during and after Covid.

A similar analysis, along with bibliometric analysis, has not been conducted before, and the study emphasizes this fact. The analysis of this study shows a parallel distribution of the three indices: Atkinson, Hoover, and Theil. Furthermore, it is suggested that the results of this study can be used to develop an index for global tourism network analysis. Similar to global tourism network analysis [9], the distribution analysis of this study can also supplement and enrich the information provided by existing tourism statistics.

The initial section of this manuscript encompasses the study's scope and a bibliometric analysis conducted using the VOSviewer program on the Web of Science database, utilizing relevant keywords associated with the research. Subsequently, the study introduces three inequality indices, namely, the Atkinson, Theil, and Hoover indices, employed to examine the distribution of tourists among the countries situated in the northern region of the Mediterranean basin from 2004 to 2020.

The analysis of the three inequality indices revealed consistent patterns. The primary hypothesis of the study posits a gradual decrease in inequality over the years, leading to a more even distribution of tourists throughout the region. While this hypothesis holds true for the majority of cases, it is important to note that there were instances of increased inequality observed in 2006, 2016, and 2020. Notably, the most substantial rise in inequality occurred in 2020, attributed to the diverse tourist policies implemented by countries in response to the 2019 pandemic. However, the increases in inequality observed in 2006 and 2016 did not demonstrate statistical significance.

In summary, this manuscript encompasses a comprehensive analysis of the distribution of tourists in the northern Mediterranean region, offering insights into the varying trends of inequality over the years and providing valuable observations regarding the impact of the Covid-19 pandemic on tourist distribution patterns.

## 2. Literature review, scope and bibliometric analysis

This section begins by defining the scope of the research, followed by a bibliometric analysis of the terms mentioned in the scope.

### 2.1 Literature review and scope of the research

The global pandemic has resulted in important and long-lasting effects in the tourism sector. There are distinctions in other economic and social crises such as the pandemic. For example, the MERS epidemic caused the loss of more than 2 million tourists in South Korea, and a corresponding loss of more than $2.5 billion in revenue [10]. In this study, the changes in the distribution of tourist numbers after 2004, during the 2008 world financial crisis and the 2019 Covid-19 crisis, are discussed.

This study focuses on analyzing the distribution of tourists among Mediterranean countries with coastlines. The analysis is conducted independently of social, political, and environmental factors within each country. Environmental factors play a crucial role in achieving sustainable coastal tourism [11]. Additionally, there are studies that have indicated the impact of terrorist activities on tourist behavior in the Mediterranean basin. For instance, tourists' decisions can be influenced by the risk of terrorism [12, 13]. There are also studies on the tourism marketing and the satisfaction of tourists [14]. Many areas such as health, education, culture [15] such as historical sites, protected areas [16], sports, beaches, mental and entertainment, and country groups with this area may have different tourist attractions. This study was carried out with an understanding of these factors in mind.

The study's scope includes the numbers of international tourists from countries bordering on the Mediterranean between 2004 and 2020. Seasonality is a significant factor in tourism, and research has been conducted to reveal its effects. Studies have demonstrated that seasonality varies significantly in the Mediterranean region, which is one of the reasons why this study focuses on this area [17]. Additionally, the availability of data for the period of interest and based on countries was a crucial factor in selecting countries with a coast in the Mediterranean basin. The study discusses the annual number of tourist arrivals to these selected countries, and the data were obtained from the World Bank repository.

Based on the 2017 Barcelona Convention Report [18] for countries with a North-coast (North countries which are geographically found above the N37-30 parallels, shown in Fig 1) to the Mediterranean basin, including Cyprus. Since the study will be based on the area of the countries, the values confirmed through the official web address of each country are taken from the world atlas website [19].

As shown in Fig 1, north side of the Mediterranean Basin is taken into account. Thirteen countries out of 21 were included in the analysis: Albania, Bosnia and Herzegovina, Croatia, Cyprus, France, Greece, Italy, Malta, Monaco, Montenegro, Slovenia, Spain, and Turkey. African countries were excluded from the analysis because their total share of global tourist arrivals is around 5%, while European countries account for over 50% [21].

Given the literature review and the significant tourist attraction of the Mediterranean basin, it became imperative to analyze the distribution of tourists within this region, specifically focusing on the European countries encompassed by the Mediterranean basin.

On the other hand, when considering the indices employed in this study and their representation in the existing literature, the Atkinson index has been particularly utilized in assessing management within the tourism sector, along with studies investigating income distribution inequality attributed to tourism [22] and studies showing the income distribution inequality caused by tourism [23].

In the examination of various indices, it is evident that the Theil index predominates within the tourism sector. In a search conducted within the journals under the web of science umbrella, findings highlight 17 studies with content related to Theil index and tourism

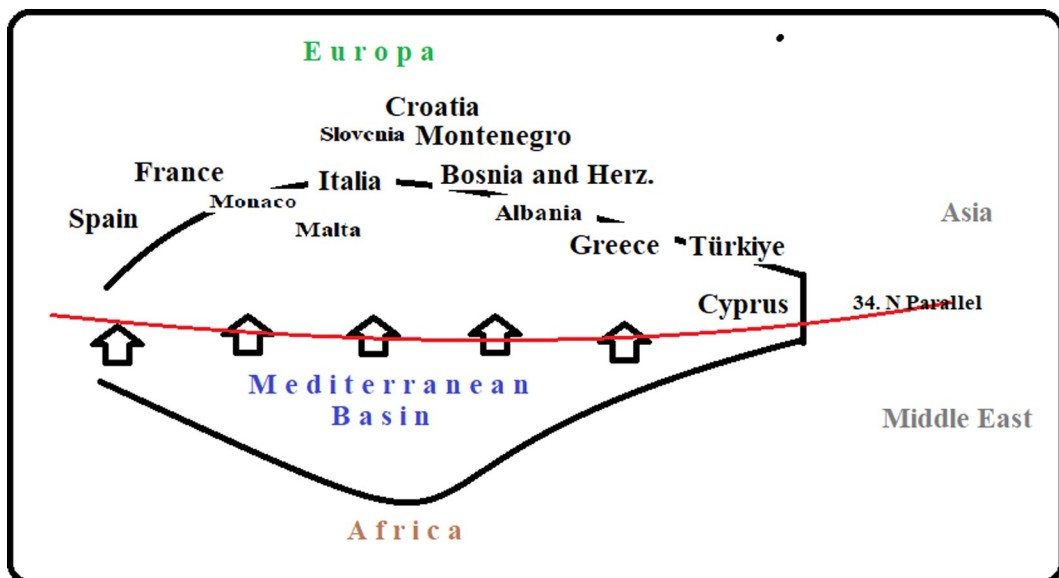

**Fig 1. Northern countries of Mediterranean Basin [20].**

keywords. TS = (Theil) Results 17 articles] Although most of them are interested in economic distribution, especially tourism demand issue [24–27].

While most of these studies center around economic distribution, notably addressing the tourism demand, there also exists research focusing on regional distribution of tourist numbers [28]. On the contrary, the Hoover index hasn't appeared in any of the studies within the web of science collection. In terms of the indices employed in this study, its distinctiveness becomes apparent, mirroring its distinct regional and research problem considerations.

## 2.2 Bibliometric analysis

Web of Science and a local Turkish research database were used to conduct the bibliometric analysis. Several queries were run in Web of Science, and the resulting data are presented in Table 1. The results of the following query were analyzed using VOSviewer.

The top 1,000 most cited results were exported from Web of Science to create a bibliographic file for use in the VOSviewer application.

(TI = (Tourists) OR TI = (Tourism)) AND (TS = (Mediterranean) OR TS = (distribution)) AND (Publication Years = 2018 or 2019 or 2020 or 2021 or 2022)

A search for inequality indexes (i.e., Theil, Hoover, or Atkinson) in the keywords of the research articles yielded only 121 results. When the term "tourism" was included in the search, no results were found.

(KP = (theil) OR KP = (atkinson) OR KP = (hoover)) AND TS = (Tourist*)

No result found [29]

The results of the query were obtained by searching the Web of Science. There are not enough studies on the subject in local databases for bibliometric analysis. The query the is stated as: (TI = (Tourists) OR TI = (Tourism)) AND (TS = (Mediterranean) OR TS = (distribution)), limited to the past five years. Out of 1,157 results, the first 1,000 were exported for

**Table 1. Search queries and results in Web of Science database.**

| Terms | With in | Query | # of Results |
|---|---|---|---|
| Tourism Tourist | Topics | (TS = (TOURISM) OR TS = (TOURIST)) | 133,252 |
| Mediterranean, Tourism Tourist | Topics | TS = (Mediterranean) AND (TS = (Tourism) OR TS = (Tourist)) | 2,204 |
| Mediterranean, Tourist | Topics | TS = (Mediterranean) AND TS = (Tourist) | 1038 |
| Inequality, Tourism Tourist | Topics | TS = (Inequality) AND (TS = (TOURISM) OR TS = (TOURIST)) | 989 |
| Atkinson, Theil, Hoover, Inequality | Title, Topics | (TI = (atkinson) OR TI = (Theil) OR TI = (Hoover)) AND TS = (Inequality) | 97 |
| Atkinson, Theil, Hoover, Inequality Index, Tourist, Tourism | Title, Topics | (TI = (atkinson) OR TI = (Theil) OR TI = ("Inequality Index") OR TI = (Hoover)) AND ((TI = (atkinson) OR TI = (Theil) OR TI = (Hoover)) AND TS = (Inequality)) AND (TS = (TOURISM) OR TS = (TOURIST)) | No results |
| Distribution Tourism Tourist | Title | (TI = (Tourists) OR TI = (Tourism)) AND TI = (distribution) | 228 |
| Mediterranean Tourist Tourism | Title | (TI = (Tourists) OR TI = (Tourism)) AND TI = (mediterranean) | 275 |
| Mediterranean, Distribution Tourist Tourism | Title | (TI = (Tourists) OR TI = (Tourism)) AND (TI = (mediterranean) OR TI = (distribution)) | 503 |
| Mediterranean, Distribution Tourist Tourism | Title, Topics | (TI = (Tourists) OR TI = (Tourism)) AND (TS = (mediterranean) OR TS = (distribution)) | 2411 |
| Tourist, Distribution | Topics | TS = ("Tourist Distribution") | 20 |
| Tourists, Tourism, Mediterranean, Distribution | Title, Topics | ((TI = (Tourists) OR TI = (Tourism)) AND (TS = (mediterranean) OR TS = (distribution)) and 2022 or 2021 or 2020 or 2019 or 2018 (Publication Years)) | 1157 |

Data obtained from [30]

further analysis, which is a logical choice as the last five years comprise half of the total results. The distribution of results over each year is presented in Table 2.

Table 2 displays the results of our study on tourism in the Mediterranean Basin, including distribution trends, from 1976 to 2022. Our findings indicate a gradual increase in tourism activity in the region over time. According to the academic journal park of Turkey, dergipark. org.tr, a search using the query "(title: tourist* OR Turi*) AND (abstract: Atkinson OR Theil OR Hoover)" resulted in 40 research papers [32]. However, none of these papers focused on the Mediterranean region [32]. Using the top 1,000 results obtained from the query, VOS-viewer was used to create network (Fig 2) and overlay (Figs 3–6) visualizations.

Fig 2 depicts the top 10 clusters, with the largest cluster being associated with the keyword "tourism". Network visualizations indicates that, within 12 different clusters of the whole key-words occurrences only three clusters include the term distribution and they are "Tourism Distribution Channels" (Cluster #1), "Income Distribution" (Cluster #5) and "distribution and spatial distribution" (Cluster #7) itself.

Fig 3 shows that, as expected, Covid-19 has been studied recently. In addition to Covid-19, spatial distribution (Fig 4) and big data (Fig 5) have been studied in the tourism sector. On the other hand, when the term "Mediterranean" is overlaid (Fig 6), it indicates that there has been no study after the second half of 2019 in the result of the web of science outputs.

As illustrated in Fig 3, the overlay visualization of our study reveals that the yellow-colored keywords represent more recent topics of investigation. Notably, the latest studies have primarily focused on Covid-19.

Fig 4 illustrates that recent studies on spatial distribution keywords and clusters do not include tourist distribution. This study thus represents a unique contribution to the research on "distribution" related keywords.

Fig 5 depicts the 12 clusters, none of which are directly related to the number of tourist distributions within the Northern Mediterranean Basin countries.

Upon magnifying the keyword "Mediterranean" as depicted in Fig 6, our analysis reveals that the most recent studies have focused on "perception", "tourist accommodation", and "stakeholders". Notably, these studies do not relate to the number of tourist distributions in the Mediterranean Basin.

**Table 2. Search for the specific query by the Web of Science.**

| Years | Number of Pub. | Years | Number of Pub. | Years | Number of Pub. |
|---|---|---|---|---|---|
| 2022 | 264 | 2009 | 74 | 1996 | 5 |
| 2021 | 233 | 2008 | 71 | 1995 | 7 |
| 2020 | 265 | 2007 | 42 | 1994 | 4 |
| 2019 | 199 | 2006 | 30 | 1993 | 2 |
| 2018 | 196 | 2005 | 27 | 1992 | 2 |
| 2017 | 171 | 2004 | 19 | 1991 | 4 |
| 2016 | 144 | 2003 | 12 | 1990 | 1 |
| 2015 | 136 | 2002 | 8 | 1989 | 1 |
| 2014 | 130 | 2001 | 8 | 1986 | 2 |
| 2013 | 124 | 2000 | 10 | 1985 | 1 |
| 2012 | 100 | 1999 | 6 | 1983 | 1 |
| 2011 | 94 | 1998 | 8 | 1981 | 1 |
| 2010 | 87 | 1997 | 2 | 1976 | 1 |

Data Obtained from: [31]

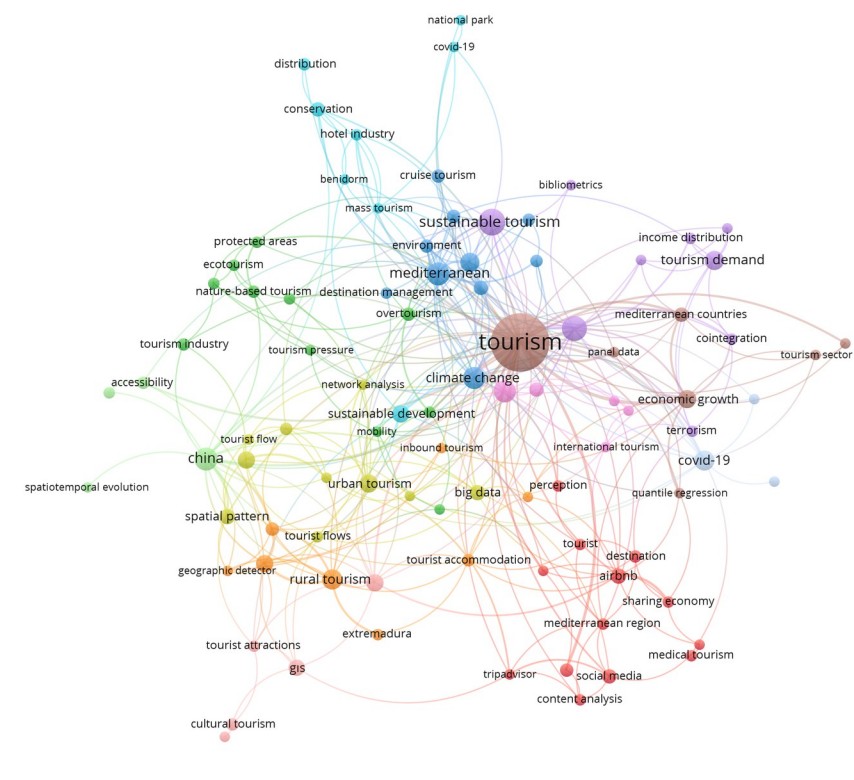

**Fig 2. Network visualization of the results.** (VOSviewer outputs).

## 3. Inequality distribution analysis of the tourist arrivals

The literature contains several studies that demonstrate the effectiveness of models that estimate regional tourism growth in forecasting future trends. These studies typically analyze historical tourism data at the country and regional level, with the aim of conducting forward-looking analyses and assessing the impact of regional or global power formations, and to inform the development of strategies for individual countries or country groups [33–35].

This study focuses on examining tourist arrival data from the World Bank database in the Mediterranean region, covering the period between 2004–2020. The data is analyzed using the Atkinson, Theil, and Hoover inequality indices to assess the degree of inequality in tourist arrivals within the region.

Table 3 demonstrates that the required data is available for all countries within the scope of our research, spanning from 2004 to 2020.

### 3.1 Atkinson Inequality Index

The Atkinson Inequality Index is used to measure the distribution of population, income, and other resources [37]. It was developed by Anthony Barnes Atkinson in 1970 [38]. A study has examined the relationship between tourism and the Atkinson Index in the Sub-Saharan Africa region and its financial distribution thresholds. However, that study is not directly related to the number of tourist arrivals for regional tourism [22].

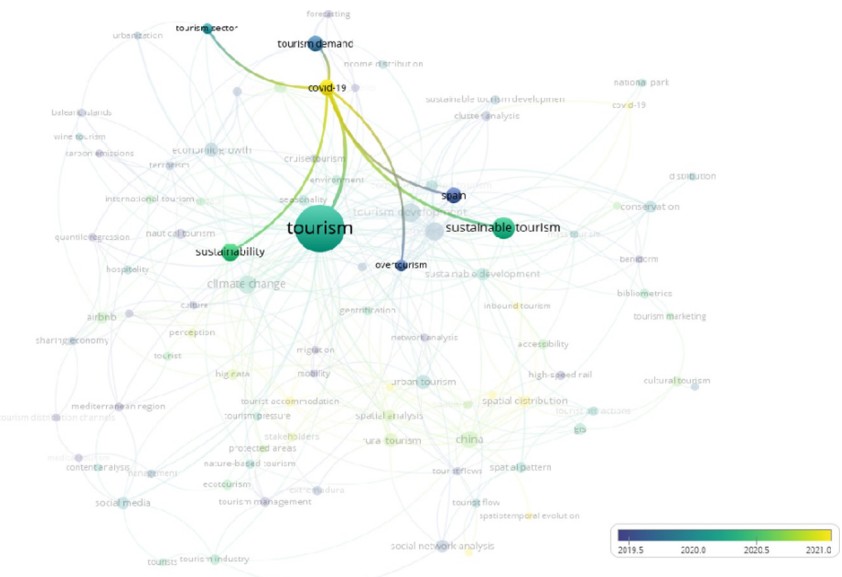

**Fig 3. Overlay visualization clusters over "Covid 19".** (VOSviewer outputs).

Atkinson inequality index is simply calculated by:

$$Atkinson\ Index\ = 1 - \left[ \sum_i^n \left( \frac{x_i}{z} \right)^{1-e} f(x_i) \right]^{\frac{1}{1-e}}$$

$x_i$: # of tourists per square km in country i

$f(x_i)$: the ratio of arrival tourists for the country i, to the total number of tourists visiting the region

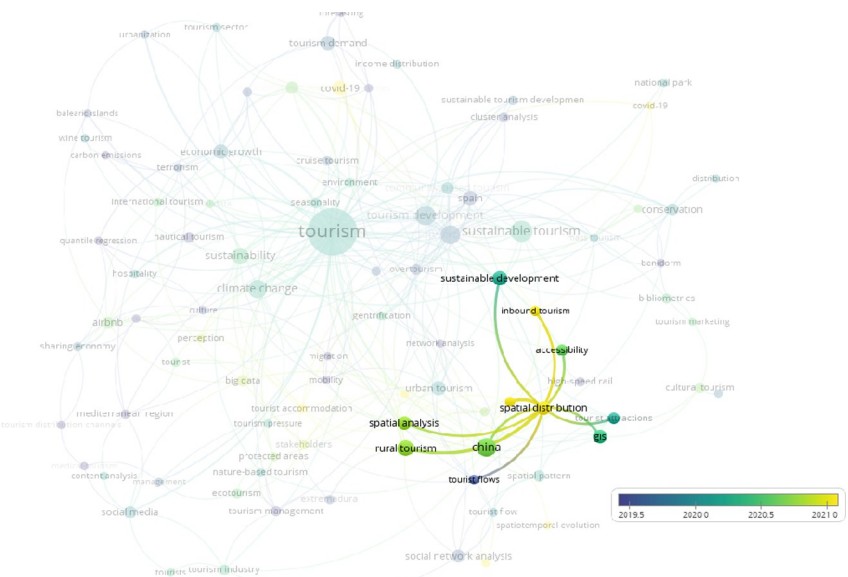

**Fig 4. Overlay visualization clusters over "spatial distribution".** (VOSviewer outputs).

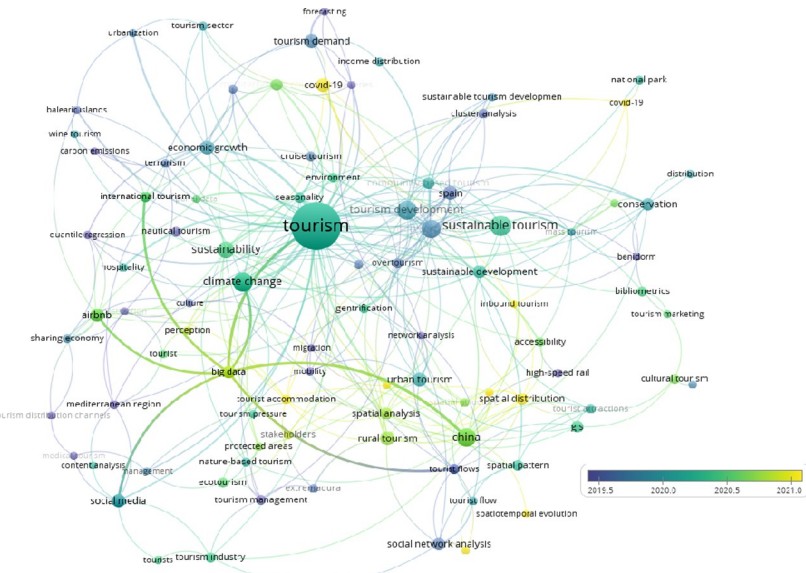

**Fig 5. Overlay visualization clusters over "big data".** (VOSviewer outputs).

z: # of tourists per square kilometer in the region

e: estimated sensitivity value (for our study it has been taken 4)

n: number of countries in the region

Calculated Atkinson Index yearly by using the data shown in the Table 4, the graph is drawn in Fig 7.

Based on the graph depicted in Fig 7, the distribution from year to year appears to follow a relatively consistent pattern, with the exception of 2016 and 2020. Atkinson Inequality Index (in %) is used to measure the distribution of the number of tourists among the countries in the

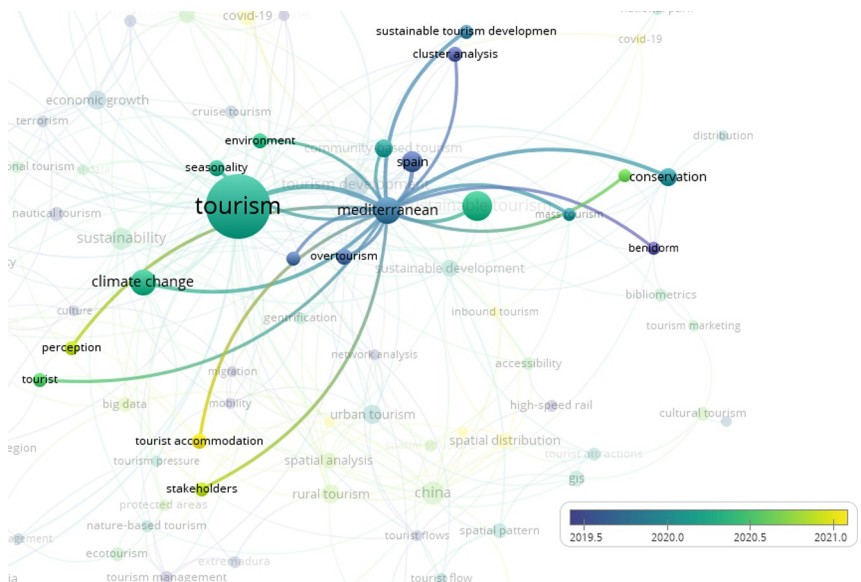

**Fig 6. Overlay visualization clusters over "Mediterranean".** (VOSviewer outputs).

**Table 3. Data source and scope: Tourist arrivals (in thousands) and country areas (in km²) for the Years 2004–2020.**

| | AL[a] | BIH | HR | CY | F | GR[b] | I | M | MC | MNE | SLO | E | TR |
|---|---|---|---|---|---|---|---|---|---|---|---|---|---|
| **Area** | 28748 | 51129 | 56594 | 9251 | 643801 | 131940 | 301338 | 316 | 2 | 13812 | 20273 | 498511 | 783562 |
| **2004** | 645 | 190 | 44974 | 2478 | 190282 | 14268 | 58480 | 1428 | 250 | 188 | 1499 | 85981 | 17517 |
| **2005** | 748 | 217 | 45762 | 2657 | 185829 | 15938 | 59230 | 1484 | 286 | 272 | 1555 | ]2563 | 21125 |
| **2006** | 937 | 256 | 47773 | 2629 | 193882 | 17284 | 66353 | 1521 | 313 | 378 | 1617 | 96152 | 19820 |
| **2007** | 1127 | 306 | 52271 | 2671 | 193319 | 17600 | 70271 | 1723 | 328 | 984 | 1751 | 98907 | 27215 |
| **2008** | 1420 | 322 | 51336 | 2370 | 193571 | 18000 | 70719 | 1832 | 324 | 1031 | 1958 | 97670 | 31138 |
| **2009** | 1856 | 311 | 47573 | 2450 | 192369 | 18400 | 71692 | 1608 | 265 | 1044 | 1824 | 91899 | 31760 |
| **2010** | 2417 | 365 | 49006 | 2626 | 189826 | 18800 | 73225 | 1815 | 279 | 1088 | 2049 | 93744 | 32997 |
| **2011** | 2932 | 392 | 46969 | 2635 | 196595 | 19200 | 75866 | 1916 | 295 | 1201 | 2236 | 99187 | 36769 |
| **2012** | 3514 | 439 | 47185 | 2626 | 197522 | 19700 | 76293 | 2007 | 292 | 1264 | 2377 | 98128 | 37715 |
| **2013** | 3256 | 529 | 48345 | 2558 | 204410 | 20112 | 76762 | 2013 | 328 | 1324 | 2502 | 103231 | 39861 |
| **2014** | 3673 | 536 | 51168 | 2780 | 206599 | 24272 | 77694 | 2162 | 329 | 1350 | 2675 | 107144 | 41627 |
| **2015** | 4131 | 678 | 55858 | 3280 | 203302 | 26114 | 81068 | 2383 | 331 | 1560 | 3022 | 109834 | 41114 |
| **2016** | 4736 | 778 | 57587 | 3286 | 203042 | 28071 | 84925 | 2592 | 336 | 1662 | 3397 | 115561 | 30907 |
| **2017** | 5118 | 923 | 59238 | 3750 | 207274 | 30161 | 89931 | 2944 | 355 | 1877 | 3991 | 121717 | 37970 |
| **2018** | 5927 | 1053 | 57668 | 4024 | 211998 | 33072 | 93228 | 3232 | 347 | 2077 | 4425 | 124456 | 46113 |
| **2019** | 6406 | 1198 | 60021 | 4117 | 217877 | 34005 | 95399 | 3519 | 363 | 2510 | 4702 | 126170 | 51747 |
| **2020** | 2658 | 197 | 21608 | 4000 | 117109 | 7406 | 38419 | 718 | 159 | 351 | 1216 | 36410 | 15971 |

Data obtained from [18, 19, 36].

[a] Plate Codes were used instead of Country Names. AL: Albania, BIH: Bosnia and Herzegovina, HY: Croatia, CY: Cyprus, F: France, GR: Greece I: Italy, M: Malta, MC: Monaco, MNE: Montenegro, SLO: Slovenia, E: Spain, TR: Türkiye.

[b] For Greece the years 2007–2012 Tourist arrivals are generated by the "linearly increase" assumptions.

**Table 4. The results of the three index Year by Year 2004–2020.** (Created by the authors).

| Years | Atkinson | Hoover | Theil |
|---|---|---|---|
| 2004 | 0,74 | 32,3 | 0,24 |
| 2005 | 0,72 | 31,2 | 0,21 |
| 2006 | 0,71 | 31,6 | 0,22 |
| 2007 | 0,67 | 30,1 | 0,19 |
| 2008 | 0,66 | 29,1 | 0,17 |
| 2009 | 0,66 | 28,6 | 0,16 |
| 2010 | 0,63 | 28,2 | 0,16 |
| 2011 | 0,62 | 27,7 | 0,15 |
| 2012 | 0,60 | 27,2 | 0,14 |
| 2013 | 0,57 | 27,2 | 0,14 |
| 2014 | 0,57 | 26,3 | 0,14 |
| 2015 | 0,54 | 26,1 | 0,14 |
| 2016 | 0,58 | 27,6 | 0,17 |
| 2017 | 0,54 | 26,5 | 0,15 |
| 2018 | 0,49 | 25,2 | 0,13 |
| 2019 | 0,47 | 24,4 | 0,12 |
| 2020 | 0,63 | 34,1 | 0,18 |

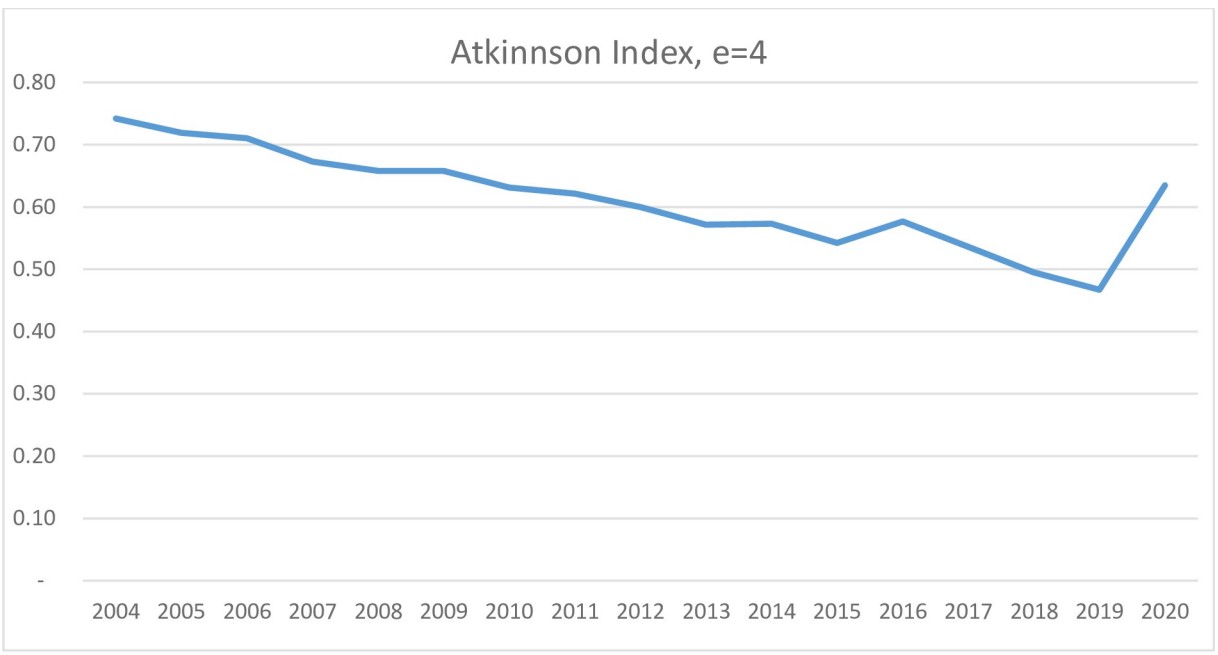

**Fig 7. Atkinson Inequality Index of the number of tourists distributed in Northern Mediterranean Basin countries from 2004 to 2020: Created by the authors.**

Northern Mediterranean Basin. The index was calculated by the authors based on data from the World Bank database.

## 3.2 Hoover Inequality Index

The Hoover Index, like other distribution indices, is one of the measures of inequality, which is created by the Edgar Malone Hoover Junior in 1936 [39]. Although it is not commonly used in the field of tourism, it is sometimes used in conjunction with other indices such as Atkinson, Theil, and Gini [40].

Hoover Index (is also known as Robin Hood Index) is calculated with the fallowing equation [41]:

$$Hoover\ Index\ = 50 \sum_{i=1}^{n} |f(x_i) - a_i|$$

$a_i$: is the ratio of the country i's area within the region.

$f(x_i)$: the ratio of arrival tourists for the country i, to the total number of tourists visiting the region.

n: number of countries in the region.

Calculated Hoover Index yearly is shown in the Table 4. The graph is shown in Fig 8.

Based on the graph presented in Fig 8, the observed changes in the variable of interest demonstrate a similar pattern to that of the Atkinson Index, characterized by alternating increases and decreases.

## 3.3 Theil (T) Inequality Index

The Theil Index, developed by Henry Theil in 1967, is a measure of economic inequality that has been widely used in various fields. Originally designed to measure losses during the

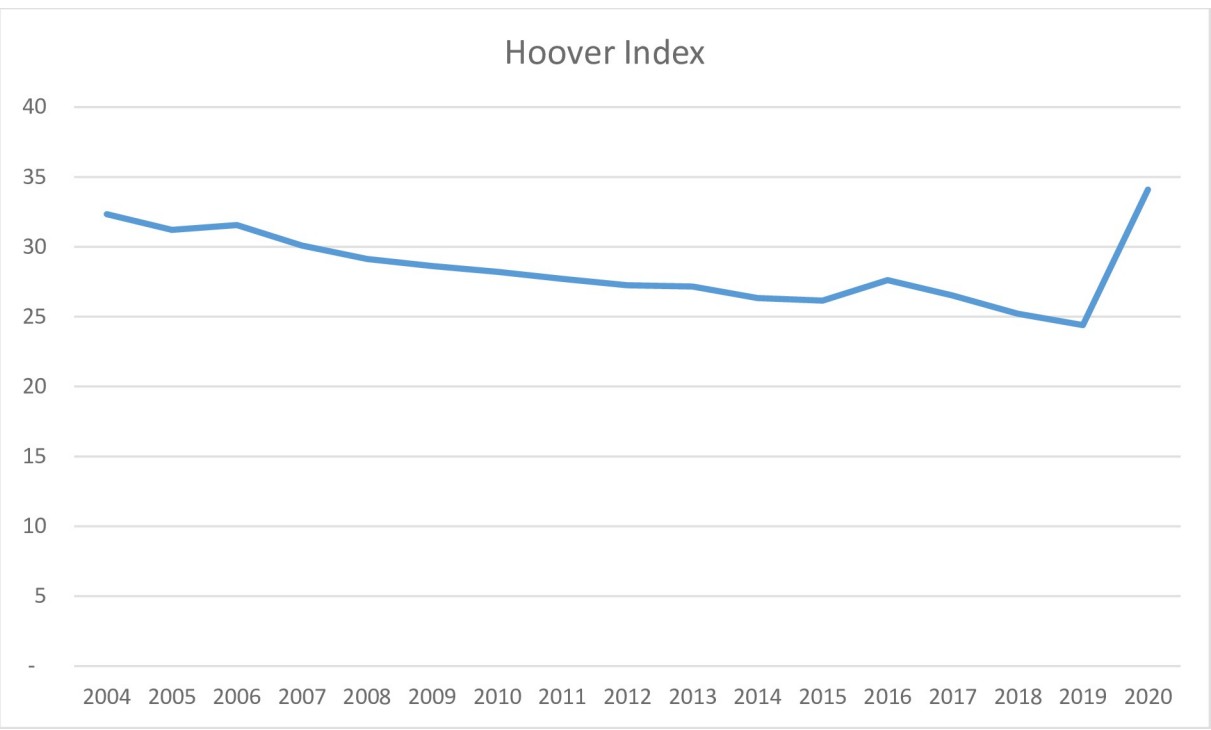

**Fig 8. Hoover Inequality Index of the number of tourists distributed among Northern Mediterranean Basin Countries, 2004–2020 (Created by the Authors).**

transfer of information between economic actors, it has since been applied to other areas such as interregional income inequality. In tourism research, the Theil Index has been used to analyze the distribution of tourist arrivals within a region or country [42]. To calculate the Theil Index, the logarithmic deviation of each region or area is calculated and then multiplied by its respective weight, which represents the size of the region or area in relation to the whole.

$$Theil\ (T)\ Index\ =\ \sum_{i=1}^{n} a_i \log \left( \frac{a_i}{f(x_i)} \right)$$

$a_i$: is the ratio of the country i's area within the region.

$f(x_i)$: the ratio of arrival tourists for the country i, to the total number of tourists visiting the region.

n: number of countries in the region.

The Theil Index has been used in various areas related to tourism, particularly in forecasting issues related to both inbound [25] and outbound tourism demand in a region. In addition, the Theil Index has been utilized for spatial and temporal comparisons in the tourism sector [43]. The yearly calculated Theil (T) Index is presented in Table 4.

As Fig 9 illustrates, the observed fluctuations in the variable of interest demonstrate a similar trend to that of the Theil index, as well as the Atkinson and Hoover indices. Specifically, the values for distribution have increased in 2016 and 2020, while decreasing in the remaining years covered by this study. This pattern may be indicative of a relatively stable distribution of tourist arrivals across the countries under investigation.

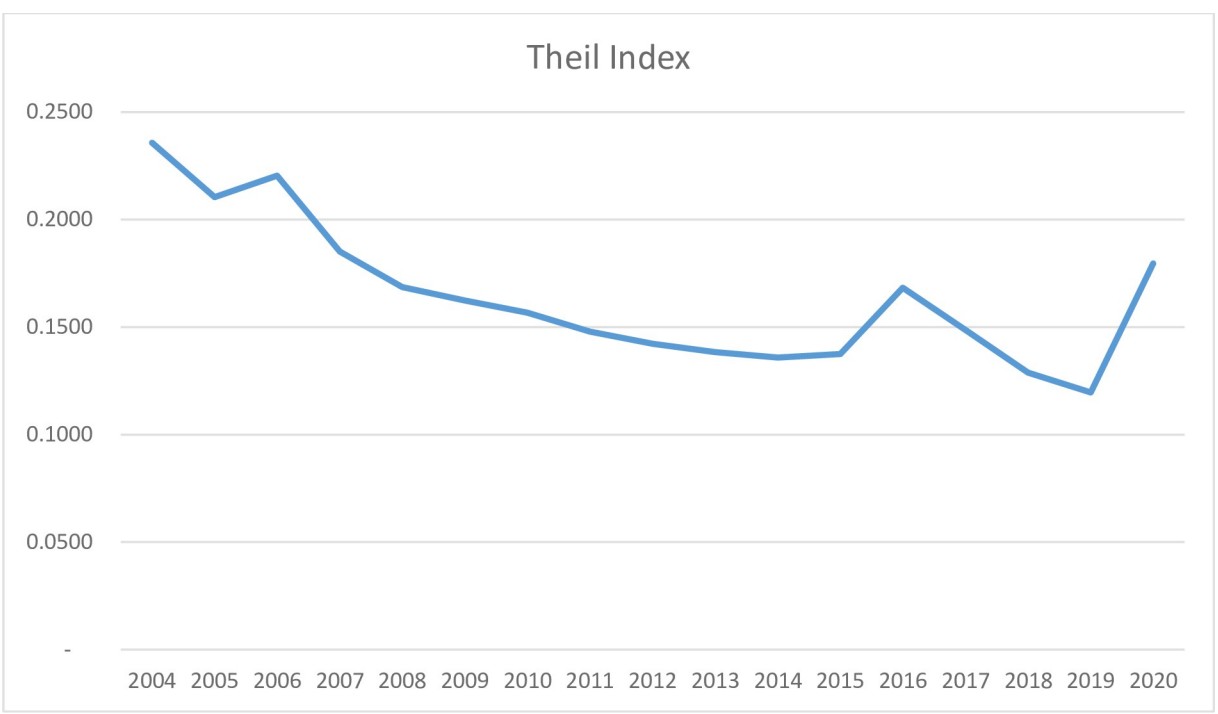

**Fig 9. Theil (T) Inequality Index of the number of tourists distributed Northern Mediterranean Basin Countries, 2004–2020.** (Created by the Authors).

## 4. Discussion

This manuscript investigates the spatial data concerning the countries within the Mediterranean basin, specifically focusing on the number of tourists visiting each country. However, it is important to note that the study does not delve into the distribution of coastal tourism; instead, it primarily presents an overview of tourist arrivals across the region. The investigation employs three distinct indices, all of which yield comparable outcomes.

An intriguing aspect that warrants further examination is the potential impact of considering per capita expenditures by tourists visiting the region. By incorporating this variable into the analysis, could we arrive at similar conclusions? One crucial assumption and constraint of this study lie in the notion that tourists, regardless of their chosen destinations within the Mediterranean basin, spend an equivalent amount on average. Thus, it is imperative to discuss the implications of this assumption on the overall findings and draw attention to any potential limitations it may impose on the study's conclusions.

On the contrary, the graphical representation of the findings reveals a noteworthy transformation, particularly in the aftermath of the pandemic and the subsequent interruption of tourism. Despite this disruption, the data indicates a continuation of the previous decreasing trend in the subsequent years. The absolute increase or decrease in tourist numbers has not been explicitly addressed; however, the pattern suggests a redistribution of total tourists among countries with a one-year delay. This observation is of significance, as it points to an ongoing trend of more equitable distribution of tourists across countries.

Moreover, the progressive evenness in the distribution of tourists assumes a critical role and should be duly considered by policymakers in the tourism sector at the national level. It is pertinent to contemplate the implementation of benchmark policies, wherein countries that

attract a relatively higher number of tourists may experience a more evenly distributed influx throughout the year, proportionate to their surface area.

The study highlights the enduring impact of the pandemic on tourism and emphasizes the importance of recognizing the evolving distribution patterns of tourists among countries. As tourism policies are formulated, the promotion of equitable distribution should be prioritized, with potential consideration of benchmark strategies to achieve a balanced and sustainable tourism sector.

The bibliometric analysis in this study focused on Dergipark as a representative of local sources, and no similar study results were observed. However, when examining the topic from the perspective of the Web of Science database, a substantial number of studies (approximately 322.000) were identified. Nevertheless, the specific keyword-based refinement resulted in a more limited pool of relevant studies, comprising only 1057 articles. Intriguingly, the analysis did not yield any findings pertaining to the measurement of inequality in the distribution of tourists within a specific region.

Considering these factors, it is justifiable to assert that the present study possesses original-ity in its research scope. From an academic standpoint, it is advisable to consider replicating the study using databases such as Scopus or others with similar comprehensive coverage.

The bibliometric investigation conducted in this study highlights the distinctiveness of its research focus, supported by the scarcity of similar findings in the selected databases. Nonethe-less, expanding the analysis to include additional databases like Scopus could potentially con-tribute further insights to the field.

An important aspect that requires discussion pertains to the absence of conclusive evidence regarding how varying scales of inequality may affect tourism types differently. It would have been beneficial to provide more concrete data concerning the distribution of beach tourism, health tourism, or sports tourism, as such information could greatly inform policymakers. However, despite the significance of this inquiry, the currently available databases do not con-tain sufficient and pertinent data to make such distinctions. To address this limitation, future studies could consider obtaining data on a country-by-country basis, allowing for a more detailed examination of the distribution patterns based on different tourism types.

In reviewing the analysis results, substantial increases in the distribution of tourists among countries in the region are evident for the years 2006, 2016, and 2020. Conversely, inequality appears to have decreased in the other years. The remarkable change observed in 2020 can be attributed to the impact of the pandemic, leading to a deterioration in inequality. However, the transformations indicating increased inequality in other years are not deemed statistically significant. It is worth noting that the pandemic-induced closures in 2020 and the resulting impact on the tourism sector have had distinct consequences for each country, leading to an overall rise in the inequality of tourism distribution due to the implemented policies.

In conclusion, the manuscript's findings highlight the need for further research to explore the potential variations in tourism distribution patterns across different types of tourism. Moreover, the impact of the pandemic on the tourism sector has been substantial, necessitat-ing comprehensive analysis to understand its effects on inequality in tourism distribution among countries in the region.

Table 4 presents the calculated values of the Atkinson, Hoover, and Theil indices, which provide measures of inequality distribution for the number of tourists in the northern Medi-terranean countries that border the Mediterranean basin between 2004 and 2020. The table provides yearly values for each index, allowing for a comprehensive analysis of inequality trends over time.

The elevated values of the Atkinson, Hoover, and Theil indices signify a lack of uniform dis-tribution among tourists visiting the region, while lower values point to a more balanced

distribution. Analyzing the combined index values presented in Table 4, spanning the years 2004 to 2020, reveals a consistent trend across all three indices. As a topic for discussion, it prompts the inquiry of whether replicating the study using other indices, like the GINI coefficient, would yield comparable results. Thus, there is potential value in conducting a parallel investigation using GINI and related metrics.

Based on the charts (Figs 7–9) and Table 4, all three indices show a parallel trend, indicating a decrease from year to year. The stability of the indices during the 2008–2009 crisis is confirmed, and a significant increase is observed due to the Covid-19 pandemic. However, the three indices indicate a slight deterioration in equality in 2016 followed by a recovery.

## 5. Conclusion

This study illustrates the distribution of the annual number of tourists among countries within the northern region (European Countries) adjacent to the Mediterranean basin from 2004 to 2020. It identifies the years when distribution homogeneity declined and the overall trend gradually shifted towards greater homogeneity, indicating a reduction in inequality.

Among the three indices, the Hoover index exhibits the highest sensitivity to the deterioration in 2019, while the Atkinson index shows the least response. To further analyze the distribution of tourists, the study could be repeated based on other factors such as the population of the country, the length of the coast in the Mediterranean, and the income level obtained from coastal tourism.

The study examines the distribution of tourists among 13 Mediterranean countries from 2004 to 2020 and finds an overall improvement in tourist distribution over the years, except for the deterioration observed in 2016 and 2020, as confirmed by the three inequality indices: Atkinson, Theil, and Hoover. The Hoover index shows the most intense deterioration in 2020, reflecting the impact of the COVID-19 pandemic on the tourism industry.

This study contributes to the literature by conducting a distribution analysis of annual tourist gains and analyzing the distribution of incoming tourists based on the country's area using inequality indices. The study's focus on the Mediterranean region, known for its significant changes in seasonality due to its importance as a tourist destination, also adds to its originality.

The study's results could be used to develop a global tourism network analysis index, similar to the one conducted by Lozano & Gutiérrez [9]. However, it is important to note that this study only examines the distribution of tourists without addressing different types of tourism. Future research could investigate the impact of different forms of tourism, such as health or nature tourism, on the distribution of tourists among countries.

This study, being novel in the literature, holds significance in illustrating the distribution of inequality indices and tourist numbers across countries. Therefore, a similar investigation using different indices is recommended. To contribute to the literature, it's crucial to evaluate diverse parameters, including income levels (per capita expenditure) influenced by tourists. Understanding the distribution and trends of tourist arrivals within a region holds crucial implications for policymakers. Moreover, it offers insights into formulating more effective policies by analyzing countries with a competitive advantage in tourism services. Conversely, it underscores the importance of investing in tourism infrastructure to ensure uniform tourist distribution in the real sectors for the future.

In conclusion, this study provides valuable insights into the distribution of tourists among Mediterranean countries and the impact of crisis events on the industry. The results suggest that analyzing the Mediterranean region based on other factors, such as the length of the coast, the number and type of hotel beds, and the volume of coastal tourism, could provide further understanding of tourist distribution in the region.

## Supporting information

**S1 Data.**
(XLS)

**S2 Data.**
(TXT)

## Author Contributions

**Conceptualization:** Sabri Öz, Cemil Can Çoktuğ.

**Data curation:** Sabri Öz.

**Formal analysis:** Sabri Öz.

**Investigation:** Sabri Öz, Pınar Başar.

**Methodology:** Sabri Öz.

**Project administration:** Adnan Veysel Ertemel, Cemil Can Çoktuğ.

**Resources:** Sabri Öz, Pınar Başar, Cemil Can Çoktuğ.

**Software:** Pınar Başar, Cemil Can Çoktuğ.

**Supervision:** Adnan Veysel Ertemel.

**Validation:** Pınar Başar.

**Visualization:** Cemil Can Çoktuğ.

**Writing – original draft:** Sabri Öz.

**Writing – review & editing:** Adnan Veysel Ertemel, Cemil Can Çoktuğ.

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
