## [Decision Letter · Decision Letter 0]

5 Jul 2023

PONE-D-23-08262Tourist Distribution In Northern Mediterranean Basin Countries: 2004-2020PLOS ONE

Dear Dr. Ertemel,

Thank you for submitting your manuscript to PLOS ONE. After careful consideration, we feel that it has merit but does not fully meet PLOS ONE’s publication criteria as it currently stands. Therefore, we invite you to submit a revised version of the manuscript that addresses the points raised during the review process.

Reviewer #1: It is a complete manuscript and fits the aims and scope of the journal’s topic.

Therefore, at least a ‘’Major Revision’’ is required to substantially improve this

manuscript. Specifically, the reviewer has the following comments:

POINT 1: The authors should include a bibliometric analysis in the title of the paper.

POINT 2: Abstract: you might try to better clarify the theoretical gap you intend to contribute to. A bit more info could have been added in terms of the used methodology and how do you analyze the data.

POINT 3: Keywords: The authors should avoid the syntax keywords, for example: ‘’distribution in tourism’’ should be divided to two keywords ‘’Tourism’’, ‘’Distribution’’.

POINT 4: A check to eliminate some typos is needed.

POINT 5: The introduction is not solid; it lacks crucial information. An introduction should be insightful and robust, providing readers with a clear understanding of the entire paper.

POINT 6: It is recommended to state the search structure in the last paragraph of the introduction.

POINT 7: The literature review is not clear.

POINT 8: The authors should justify why choosing WOS rather than Scopus for bibliometric analysis.

POINT 9: The authors have neglected many studies conducted in distribution in tourism by using these keywords.

POINT 10: The layout is missed in Figures 2 and 3.

POINT 11: The methodology is not well described and justified.

POINT 12: The format of references is not adequate with PlOS ONE requirement.

Reviewer #2: Innovation potential is very limited

Research hypothesis is not clearly formulated

What authors aim to analyze and conclude is not convincingly demonstrated and formulated

The paper although emphasis in Tourist sector does not provide and useful information and the state of the art (introduction) does not highlight the importance of hospitality industry. There are too many references highlighting the importance of Mediterranean area and why is chosen from other destinations and this is not explained

In addition, methodological approach does not include several elements e.g how the data were collected, how the review was performed, etc. Hence, i couldn't find any statistical analysis or wasn't so clear

results provide typical numbers without any explanation of the importance of the specific KPIs that Authors used.

The paper lack discussion section in depth

Reviewer #3: The paper topic is interesting. I have some comments, which must be addreesed before the paper is considered for publication.

1- the authors used WoS and a Turkish database. However, it's worth noting that the adequacy of resources for a bibliometric analysis depends on the specific research question, scope, and objectives of your study. In some cases, researchers may also consider other databases like Scopus, Google Scholar, or specific subject-specific databases to ensure a more comprehensive analysis. It's essential to choose the most appropriate databases and sources based on your research needs and disciplinary context. Please justify why you relied only on these databases.

2- the authors mentioned that "This study contributes to the literature by conducting a distribution analysis of annual

tourist gains and analyzing the distribution of incoming tourists based on the country's area

using inequality indices"

However, what does that mean to the existing literature is not discussed.

3- There are no theoretical or empirical implications sections to discuss what the findings means to the literature or practitioners. How the findings will benefit other than helping in developing global network analysis index.

4- in the scope section:

There should be more discussion tot he existing literature to know the conceptual background of the topic.

5- There should be either a research context to show data about the northern mideternian countries tourists or include these data in the introduction to make this context familiar to reader from outside the area. Check the below reference for guidance on the research context:

Mahrous, A. A. (2016). Implications of the use of social media for pre-purchase information searches for automobiles. International Journal of Technology Marketing, 11(3), 254-275.

We look forward to receiving your revised manuscript.

Kind regards,

Wilmer Carvache-Franco, PhD.

Academic Editor

PLOS ONE

Journal Requirements:

2. We note that Figure 1 in your submission contain [map/satellite] images which may be copyrighted. All PLOS content is published under the Creative Commons Attribution License (CC BY 4.0), which means that the manuscript, images, and Supporting Information files will be freely available online, and any third party is permitted to access, download, copy, distribute, and use these materials in any way, even commercially, with proper attribution. For these reasons, we cannot publish previously copyrighted maps or satellite images created using proprietary data, such as Google software (Google Maps, Street View, and Earth). For more information, see our copyright guidelines: http://journals.plos.org/plosone/s/licenses-and-copyright.

In the figure caption of the copyrighted figure, please include the following text: “Reprinted from [ref] under a CC BY license, with permission from [name of publisher], original copyright [original copyright year].

Additional Editor Comments:

Dear authors, please make the corrections provided by your reviewers.

Reviewers' comments:

Reviewer's Responses to Questions

**Comments to the Author**

1. Is the manuscript technically sound, and do the data support the conclusions?

Reviewer #1: Partly

Reviewer #2: No

Reviewer #3: Partly

2. Has the statistical analysis been performed appropriately and rigorously? 

Reviewer #1: Yes

Reviewer #2: No

Reviewer #3: No

3. Have the authors made all data underlying the findings in their manuscript fully available?

Reviewer #1: Yes

Reviewer #2: No

Reviewer #3: No

4. Is the manuscript presented in an intelligible fashion and written in standard English?

Reviewer #1: Yes

Reviewer #2: No

Reviewer #3: Yes

5. Review Comments to the Author

Reviewer #1: It is a complete manuscript and fits the aims and scope of the journal’s topic.

Therefore, at least a ‘’Major Revision’’ is required to substantially improve this

manuscript. Specifically, the reviewer has the following comments:

POINT 1: The authors should include a bibliometric analysis in the title of the paper.

POINT 2: Abstract: you might try to better clarify the theoretical gap you intend to contribute to. A bit more info could have been added in terms of the used methodology and how do you analyze the data.

POINT 3: Keywords: The authors should avoid the syntax keywords, for example: ‘’distribution in tourism’’ should be divided to two keywords ‘’Tourism’’, ‘’Distribution’’.

POINT 4: A check to eliminate some typos is needed.

POINT 5: The introduction is not solid; it lacks crucial information. An introduction should be insightful and robust, providing readers with a clear understanding of the entire paper.

POINT 6: It is recommended to state the search structure in the last paragraph of the introduction.

POINT 7: The literature review is not clear.

POINT 8: The authors should justify why choosing WOS rather than Scopus for bibliometric analysis.

POINT 9: The authors have neglected many studies conducted in distribution in tourism by using these keywords.

POINT 10: The layout is missed in Figures 2 and 3.

POINT 11: The methodology is not well described and justified.

POINT 12: The format of references is not adequate with PlOS ONE requirement.

Reviewer #2: Innovation potential is very limited

Research hypothesis is not clearly formulated

What authors aim to analyze and conclude is not convincingly demonstrated and formulated

The paper although emphasis in Tourist sector does not provide and useful information and the state of the art (introduction) does not highlight the importance of hospitality industry. There are too many references highlighting the importance of Mediterranean area and why is chosen from other destinations and this is not explained

In addition, methodological approach does not include several elements e.g how the data were collected, how the review was performed, etc. Hence, i couldn't find any statistical analysis or wasn't so clear

results provide typical numbers without any explanation of the importance of the specific KPIs that Authors used.

The paper lack discussion section in depth

Reviewer #3: The paper topic is interesting. I have some comments, which must be addreesed before the paper is considered for publication.

1- the authors used WoS and a Turkish database. However, it's worth noting that the adequacy of resources for a bibliometric analysis depends on the specific research question, scope, and objectives of your study. In some cases, researchers may also consider other databases like Scopus, Google Scholar, or specific subject-specific databases to ensure a more comprehensive analysis. It's essential to choose the most appropriate databases and sources based on your research needs and disciplinary context. Please justify why you relied only on these databases.

2- the authors mentioned that "This study contributes to the literature by conducting a distribution analysis of annual

tourist gains and analyzing the distribution of incoming tourists based on the country's area

using inequality indices"

However, what does that mean to the existing literature is not discussed.

3- There are no theoretical or empirical implications sections to discuss what the findings means to the literature or practitioners. How the findings will benefit other than helping in developing global network analysis index.

4- in the scope section:

There should be more discussion tot he existing literature to know the conceptual background of the topic.

5- There should be either a research context to show data about the northern mideternian countries tourists or include these data in the introduction to make this context familiar to reader from outside the area. Check the below reference for guidance on the research context:

Mahrous, A. A. (2016). Implications of the use of social media for pre-purchase information searches for automobiles. International Journal of Technology Marketing, 11(3), 254-275.

6. PLOS authors have the option to publish the peer review history of their article (what does this mean?). If published, this will include your full peer review and any attached files.

Reviewer #1: No

Reviewer #2: No

Reviewer #3: No

---

## [Author Response · Author response to Decision Letter 0]

5 Sep 2023

Reviewer #1: 

POINT 1: The authors should include a bibliometric analysis in the title of the paper.

Thank you for your feedback. 

The present paper does not only make a bibliometric analysis but rather used a mix methodology including bibliometric analysis and three distinct inequality index, namely; Atkinson, Hoover and Theil index. As such not including *bibliometric analysis* in the title makes more sense.

POINT 2: Abstract: you might try to better clarify the theoretical gap you intend to contribute to. A bit more info could have been added in terms of the used methodology and how do you analyze the data.

To our knowledge, the study as it is the first to analyze the distribution of tourists considering the Mediterranean using the Atkinson, Theil, and Hoover inequality indices and also being supported by bibliometric analysis. This is pointed out in the “originality” section of the abstract

POINT 3: Keywords: The authors should avoid the syntax keywords, for example: ‘’distribution in tourism’’ should be divided to two keywords ‘’Tourism’’, ‘’Distribution’’.

Thank you for the feedback. We have corrected the keywords section accordingly.

POINT 4: A check to eliminate some typos is needed. 

Thank you for the comment. The whole text has been proofread again to correct some typos. 

POINT 5: The introduction is not solid; it lacks crucial information. An introduction should be insightful and robust, providing readers with a clear understanding of the entire paper.

Thank you for the feedback. In the light of reviewer comments, the introduction part has been substantially expanded. The new and improved paragraphs can be seen with track changes option which are paragraphs # 2,4,5,6,7,8,10,11,12.

POINT 6: It is recommended to state the search structure in the last paragraph of the introduction.

Thank you for the recommendation. The recommended text is added to near the end of the introduction part.

POINT 7: The literature review is not clear.

The manuscript has been substantially expanded to encompass up-to-date literature on the subject. 

POINT 8: The authors should justify why choosing WOS rather than Scopus for bibliometric analysis.

WOS was chosen as it is one of the world’s premier databases for published articles and citations and includes publications in top-tier journals and is most suited for literature review (Davaci & Gunkel, 2023). WoS is the most suitable database for data mining and has become one of the primary databases used by scholars for conducting bibliometric analysis (Forliano et al. 2021). Besides, a vast majority of top-tier journals are indexed in WOS and Scopus simultaneously. Therefore, only the WOS database was chosen to conduct the bibliometric analysis of the study. 

POINT 9: The authors have neglected many studies conducted in distribution in tourism by using these keywords.

Although there are many studies on tourism using each of inequality indices (Theil, Atkinson and Hoover), there is a shortage of studies measuring inequalities based on the number of tourist arrivals between Mediterranean basin countries. This point is elaborated in the discussion section.

POINT 10: The layout is missed in Figures 2 and 3.

As stated in the PLoS One guidelines, the figures were removed from the manuscript text file and put to the submission as separate files. 

POINT 11: The methodology is not well described and justified.

Thank you for the feedback. Based on the recommendation, the methodology is described and justified in the introduction part.

POINT 12: The format of references is not adequate with PLOS ONE requirement.

The references are thoroughly revised to correct any typos and mistakes.

Reviewer #2: Innovation potential is very limited

Thank you for the feedback. The novelty of the study is highlighted in the introduction and conclusion parts.

Research hypothesis is not clearly formulated

Thank you for the feedback. Based on the recommendation, research hypothesis is stated in the last part of the introduction

What authors aim to analyze and conclude is not convincingly demonstrated and formulated

The paper, although emphasis in the Tourist sector does not provide and useful information and the state of the art (introduction) does not highlight the importance of hospitality industry. There are too many references highlighting the importance of Mediterranean area and why is chosen from other destinations and this is not explained

In the light of the recommendation, new text has been added that highlight the importance of the hospitality industry and specifically the Mediterranean area along with accompanying references

In addition, methodological approach does not include several elements e.g how the data were collected, how the review was performed, etc. Hence, i couldn't find any statistical analysis or wasn't so clear

Thank you for the recommendation. The requested parts can be seen in the introduction (lines 67-72) as below:

The number of tourists was analyzed based on the land size of the countries, and the study was repeated using three inequality indices: Atkinson, Hoover, and Theil. The regional distribution of tourists was discussed using data obtained from the World Bank database between 2004 and 2020. The dataset used in this study is derived from the arrivals of tourists in the respective countries.

results provide typical numbers without any explanation of the importance of the specific KPIs that Authors used.

Thank you for the recommendation. Based on the reviewer’s feedback, explanation of the importance of the KPIs were added to the accompanying section.

The paper lack discussion section in depth

Thank you for the recommendation. In light of the comments, a detailed discussion section is added to the manuscript 

Reviewer #3: The paper topic is interesting. I have some comments, which must be addreesed before the paper is considered for publication.

1- the authors used WoS and a Turkish database. However, it's worth noting that the adequacy of resources for a bibliometric analysis depends on the specific research question, scope, and objectives of your study. In some cases, researchers may also consider other databases like Scopus, Google Scholar, or specific subject-specific databases to ensure a more comprehensive analysis. It's essential to choose the most appropriate databases and sources based on your research needs and disciplinary context. Please justify why you relied only on these databases.

WOS was chosen as it is one of the world’s premier databases for published articles and citations and includes publications in top-tier journals and is most suited for literature review (Davaci & Gunkel, 2023). WoS is the most suitable database for data mining and has become one of the primary databases used by scholars for conducting bibliometric analysis (Forliano et al. 2021). Besides, a vast majority of top-tier journals are indexed in WOS and Scopus simultaneously. Therefore, only the WOS database was chosen to conduct the bibliometric analysis of the study. 

2- the authors mentioned that "This study contributes to the literature by conducting a distribution analysis of annual

tourist gains and analyzing the distribution of incoming tourists based on the country's area

using inequality indices"

However, what does that mean to the existing literature is not discussed.

Thank you for the feedback. Implications of the study on existing literature is elaborated more in the discussion section

3- There are no theoretical or empirical implications sections to discuss what the findings means to the literature or practitioners. How the findings will benefit other than helping in developing global network analysis index.

Thank you for the feedback. In the light of the recommendation, theoretical and managerial implications were added near the end of the conclusion part.

4- in the scope section:

There should be more discussion to the existing literature to know the conceptual background of the topic.

In the light of the recommendation, a number of other related studies were discussed and cited in the introduction and literature review part of the manuscript. 

5- There should be either a research context to show data about the northern mideternian countries tourists or include these data in the introduction to make this context familiar to reader from outside the area. Check the below reference for guidance on the research context:

Mahrous, A. A. (2016). Implications of the use of social media for pre-purchase information searches for automobiles. International Journal of Technology Marketing, 11(3), 254-275.

Thank you for the feedback. Based on the recommendation, research context is elaborated in more detail in the introduction part.

---

## [Decision Letter · Decision Letter 1]

18 Oct 2023

Tourist Distribution In Northern Mediterranean Basin Countries: 2004-2020

PONE-D-23-08262R1

Dear Dr. Ertemel,

We’re pleased to inform you that your manuscript has been judged scientifically suitable for publication and will be formally accepted for publication once it meets all outstanding technical requirements.

Kind regards,

Wilmer Carvache-Franco, PhD.

Academic Editor

PLOS ONE

Additional Editor Comments (optional):

Reviewers' comments:

Reviewer's Responses to Questions

**Comments to the Author**

1. If the authors have adequately addressed your comments raised in a previous round of review and you feel that this manuscript is now acceptable for publication, you may indicate that here to bypass the “Comments to the Author” section, enter your conflict of interest statement in the “Confidential to Editor” section, and submit your "Accept" recommendation.

Reviewer #1: All comments have been addressed

2. Is the manuscript technically sound, and do the data support the conclusions?

Reviewer #1: Yes

3. Has the statistical analysis been performed appropriately and rigorously? 

Reviewer #1: Yes

4. Have the authors made all data underlying the findings in their manuscript fully available?

Reviewer #1: Yes

5. Is the manuscript presented in an intelligible fashion and written in standard English?

Reviewer #1: Yes

6. Review Comments to the Author

Reviewer #1: The paper, as it exists in its present form, has met the necessary criteria and standards for publication, indicating that it is ready to be published without requiring any further revisions or modifications

7. PLOS authors have the option to publish the peer review history of their article (what does this mean?). If published, this will include your full peer review and any attached files.

Reviewer #1: No

---

## [Editor Report · Acceptance letter]

25 Oct 2023

PONE-D-23-08262R1 

Tourist Distribution in Northern Mediterranean Basin Countries: 2004-2020 

Dear Dr. Ertemel:

I'm pleased to inform you that your manuscript has been deemed suitable for publication in PLOS ONE. Congratulations! Your manuscript is now with our production department. 

Kind regards, 

on behalf of

Dr. Wilmer Carvache-Franco 

Academic Editor

PLOS ONE